# Effective Extraction of the Al Element from Secondary Aluminum Dross Using a Combined Dry Pressing and Alkaline Roasting Process

**DOI:** 10.3390/ma15165686

**Published:** 2022-08-18

**Authors:** Han Lv, Mingzhuang Xie, Zegang Wu, Lili Li, Runjie Yang, Jinshan Han, Fengqin Liu, Hongliang Zhao

**Affiliations:** 1State Key Laboratory of Advanced Metallurgy, University of Science and Technology Beijing, Beijing 100083, China; 2School of Metallurgical and Ecological Engineering, University of Science and Technology Beijing, Beijing 100083, China

**Keywords:** aluminum dross, recycling, thermodynamic analysis, kinetics

## Abstract

Secondary aluminum dross (SAD) is a hazardous solid waste discharged from aluminum electrolysis and processing and the secondary aluminum industries, which causes severe environmental pollution and public health disasters. The stable presence of the α-Al_2_O_3_ and MgAl_2_O_4_ phases in SAD makes it difficult for it to be efficiently utilized. A combined dry pressing and alkaline roasting process was proposed for extracting the valuable Al element from SAD. Two alkaline additives (NaOH and Na_2_CO_3_) were selected as a sodium source for extracting the aluminum source from SAD in order to perform the thermodynamic analysis and roasting experiments. The phase transition behavior and the leaching performance tests were conducted using X-ray diffraction, scanning electron microscopy, X-ray fluorescence, leaching kinetics and thermal analysis. The recovery of Al and Na reached the values of 90.79% and 92.03%, respectively, under the optimal conditions (roasting temperature of 1150 °C, Na_2_CO_3_/Al_2_O_3_ molar ratio of 1.3, roasting time of 1 h, leaching temperature of 90 °C, L/S ratio of 10 mL·g^−1^ and leaching time of 30 min). Meanwhile, the removal efficiency of N and Cl reached 98.93% and 97.14%, respectively. The leaching kinetics indicated that the dissolution of NaAlO_2_ clinkers was a first-order reaction and controlled by layer diffusion process. The green detoxification and effective extraction of the Al element from SAD were simultaneously achieved without any pretreatments.

## 1. Introduction

With the rapid development of the global aluminum industry, the accumulation of hazardous solid wastes has become a serious environmental problem. Aluminum dross (AD) is a hazardous solid waste discharged during the electrolysis, processing and regeneration of aluminum, which can usually be divided into primary aluminum dross (PAD) and secondary aluminum dross (SAD). Primary aluminum dross (PAD) is generated in primary smelters and contains 15–80% metallic aluminum [1]. On the other hand, SAD is a by-product of PAD produced after mechanical screening or remelting with flux to recover the metallic aluminum, and it contains less than 10% aluminum [2]. Currently, the recovery process for extracting metallic aluminum from PAD is comparatively mature and has been applied in most of the aluminum processing plants [3]. The composition of SAD depends on the sources of aluminum scrap, the remelting technology, the type of additives, and the employed process. Generally, SAD consists of alumina (Al_2_O_3_), aluminum nitride (AlN), metallic aluminum, magnesium aluminum spinel (MgAl_2_O_4_), other impurity oxides and salts [4]. Globally, the aluminum industry generates more than 3 million tons of SAD every year. Around 95% of the SAD is stockpiled in landfill sites due to its complex composition and the technical difficulties involved in processing it [5].

The toxic and hazardous substances in aluminum dross are mainly AlN and chloride salts. Aluminum nitride (AlN) is produced due to the reaction of molten aluminum with nitrogen during the formation of AD, which reacts very easily with water and moist air to release toxic and irritating ammonia [6]. One ton of SAD is capable of releasing approximately 109 m^3^ of ammonia, which can burn the skin and the respiratory tract and even cause serious pulmonary diseases [7]. Ammonia also reacts with acidic gases in the atmosphere to produce aerosols of ammonium salts, posing severe air pollution risks. The arbitrary stockpiling of SAD can lead to the infiltration of soluble chlorides into the soil and groundwater, posing a serious environmental threat. Additionally, as the particles of SAD are extremely fine, they are easily dispersed in the air during processing, transportation and landfill, causing silicosis and bronchitis through excessive inhalation [8]. Therefore, SAD is considered as a hazardous solid waste in most of the countries [9].

The development of more sustainable processes to mitigate environmental pollution is absolutely essential. Considering hazardous wastes as alternative materials for preparing valuable products is a potential method for reducing the generation and management of wastes [10]. SAD is an excellent alternative resource for the development of the alumina industry, especially in view of the increasing alumina production and the growing scarcity of bauxite [11]. However, due to the complex composition of SAD and technical limitations, it is difficult to effectively extract Al elements from SAD.

In recent years, scholars have aimed to develop a variety of methods for the efficient recovery of Al element from SAD under different extraction conditions [12,13]. Some of them have achieved good results. However, there is no single method that can simultaneously achieve high Al extraction efficiency from SAD in a sustainable process [14]. Therefore, the large-scale utilization of SAD is severely restricted, and most studies are still conducted at the laboratory scale [15]. Current studies on the extraction of Al from SAD mainly focus on two processing routes: pyro-metallurgical and hydro-metallurgical processes [16]. The pyro-metallurgical processes mainly involve the alkaline roasting process with a leaching pretreatment and the plasma arc melting process. High Al extraction efficiency and high purity products can be achieved using the former process. However, this process requires a leaching pretreatment for the desalination and denitrification, which produces toxic ammonia and salt-containing waste liquid, thus greatly increasing the recovery costs [17]. The extremely high processing temperatures and energy consumption of the latter process make it unattractive for large-scale applications. The hydro-metallurgical routes are generally carried out by acidic or alkali leaching [18,19].

The acid leaching causes the impurity ions (Mg^2+^, Fe^3+^, Ca^2+^, NH^4+^ and Cl^−^) in SAD to enter the leaching solution, which makes it difficult to separate them from valuable aluminum ions and synthesize high-purity products [20,21]. Moreover, the alkaline leaching can yield high-purity NaAlO_2_ solutions for the synthesis of high-value products; however, the recovery of Al obtained under either atmospheric or high pressure is too low [22]. The main problem with hydro-metallurgical routes is that a large amount of the stable α-Al_2_O_3_ and MgAl_2_O_4_ phases in SAD is difficult to dissolve in acid or alkali solution, resulting in the low recovery of Al [23,24]. Therefore, the key to developing a sustainable process for recycling SAD into a valuable product lies in the efficient recovery of the Al element [25,26].

In the present study, a novel promising method comprising of dry pressing and alkaline roasting is developed, as shown in Figure 1. The proposed process aims at economically achieving high Al recovery.

## 2. Materials and Methods

The process presented in Figure 1 consists of three stages of dry molding, roasting and alkaline leaching. First, according to the content of Al element, SAD was mixed with a certain amount of alkali. The mixture was dry pressed under a constant pressure to produce cylindrical samples. Then, the clinkers were roasted at a high temperature and leached in alkaline solution to obtain NaAlO_2_ solution, which can be used to produce alumina. During the roasting process, AlN was decomposed into harmless N_2_, while the chlorate was separated and recovered after volatilization into the gaseous phase.

First, the thermodynamic analysis was carried out. Then, the roasting and leaching experiments were conducted to investigate the phase transition and to optimize the recovery efficiency. The leaching kinetics and the apparent activation energy of the NaAlO_2_ leaching process were discussed. The developed extraction process described here has several innovations, some of which are as follows:

(1) The roasting process can achieve the sustainable treatment and extraction of the valuable Al element from SAD without any pretreatment. (2) Dry pressing was applied to avoid the generation of harmful ammonia and liquid waste and to decrease the energy consumption of the roasting process. (3) The reaction exhibited a high yield and was simple and sustainable. (4) The chlorate in the SAD was volatilized in the roasting process and could be recovered after condensation.

### 2.1. Materials

The SAD used in this study came from an aluminum remelting plant in Guangxi, China. The X-ray diffraction (XRD) analysis of the SAD showed that the aluminum-containing phases included the metallic aluminum, α-Al_2_O_3_, γ-Al_2_O_3_, AlN and MgAl_2_O_4_, as shown in Figure 2. These were the products of the reactions of the molten aluminum with the O_2_, N_2_ and Mg impurities during the formation and remelting of the SAD. Additionally, some impurity oxides (Fe_2_O_3_ and SiO_2_) and salts, including chlorides (NaCl) and fluorides (CaF_2_), were present in the SAD. The salts were added as a flux during the melting process to facilitate the heat transfer and reduce the oxidation of the molten aluminum. The chemical analysis of the SAD is presented in Table 1. Nearly 40 wt.% of the sample was made up of Al element, indicating that the SAD was a solid waste with a high utilization value. All the nitrogen was transformed into the phase of AlN with a content of 19.62 wt.%.

### 2.2. Experimental Procedure

Figure 3 illustrates the schematic of the experimental setup. First, 100 g of SAD was accurately weighed and mixed thoroughly with a certain amount of alkali. In order to produce the specific compounds during the roasting process, the formulation of the SAD and alkali was controlled according to the molar ratio of Na_2_O to Al_2_O_3_ (n(N/A)). The mixtures were dry pressed at 30 MPa to produce cylindrical samples with a diameter and length of 20 mm and 40 mm, respectively. The cylindrical samples were placed in a corundum crucible with dimensions of 120 mm × 80 mm × 30 mm and roasted in a muffle furnace under an atmospheric environment at a preset temperature for 1 h, with a constant heating rate of 10 °C min^−1^. The crucible was taken out and placed in a desiccator to cool to room temperature. The roasting clinkers were crushed to 200 mesh in a mortar and leached in caustic liquor to obtain NaAlO_2_ solution. The leaching residue was filtered and rinsed several times with boiling water. According to the industrial dissolution conditions of bauxite clinkers, the caustic concentration of the leaching solution was maintained at 60 g·L^−1^.

### 2.3. Characterization Methods

The Gibbs free energy of the chemical reactions that can occur during roasting was analyzed using the reaction module of the software package Factsage (ver. 7.0, Thermfact, Montreal, Canada). The contents of Al and Na in the roasting clinkers and leaching residues were determined using X-ray fluorescence (XRF), and the recovery efficiencies of the Al and Na were calculated using Equations (1) and (2), respectively:(1)$$\eta_{\mathrm{Al}}{=[\mathrm{Al}}_{1}{\;-\;\mathrm{Al}}_{2}{(\mathrm{Mg}}_{1}{/\mathrm{Mg}}_{2}{)/\mathrm{Al}}_{1}$$
(2)$$\eta_{\mathrm{Na}}{=[\mathrm{Na}}_{1}{\;-\;\mathrm{Na}}_{2}{(\mathrm{Mg}}_{1}{/\mathrm{Mg}}_{2}{)/\mathrm{Na}}_{1}$$
where ${\mathrm{Al}}_{1}$, ${\mathrm{Mg}}_{1}$ and ${\mathrm{Na}}_{1}$ are the contents of Al, Mg and Na in the roasting clinkers, respectively, wt.%; and ${\mathrm{Al}}_{2}$, ${\mathrm{Mg}}_{2}$ and ${\mathrm{Na}}_{2}$ are the contents of Al, Mg and Na in the leaching residues, respectively, wt.%. The content of soluble chlorate in the SAD samples was determined using a chloride ion activity meter (PCL-202, INESA Scientific Instruments, Shanghai, China). The content of AlN in the SAD samples was determined using the Kjeldahl method [27].

For the mineralogical study, an X-ray diffractometer (PW1710, Philips, Amsterdam, The Netherlands) was used with Cu Kα-radiation at 40.0 kV and 30.0 mA. The XRD tests were conducted within the 2θ range of 10–60° using a scanning speed of 0.1°·min^−1^. Before testing, the samples were crushed to pass through a 200-mesh sieve and dried in an oven at 105 °C for 2 h. The diffractograms were identified with the help of the ICDD (International Centre for Diffraction Data) Powder Diffraction File (PDF-2) reference database and the Jade (vs. 6.0). The chemical composition of SAD was determined by XRF (mAX, AXIOS, Alamelo, The Netherlands). Micrographs of the roasting clinkers and leaching residues were observed using a scanning electron microscope (Regulus 8100, Hitachi, Japan) at an accelerating voltage of 15.0 kV. The microscopy samples were crushed to pass through a 200-mesh sieve and dried in an oven at 105 °C for 2 h, then sprayed evenly on the conductive adhesive and metal-lized with carbon in a vacuum evaporator (Q150T ES, Quorum, East Grinstead, Britain). An EDS (ul-tra-DLD, Shimadzu, Kyoto, Japan) connected with SEM was used to perform elemental anal-ysis on the sample particles. The elements with the content of 3%~20 wt.% have the accuracy of relative error <10%. And the elements with content greater than 20 wt.% have the higher accuracy of relative error <5%.

## 3. Results and Discussion

### 3.1. Thermodynamic Analysis

The amounts of the reactants and products under different roasting conditions were predicted using thermodynamic analysis. Two alkalis (NaOH and Na_2_CO_3_) were selected as additives to perform the thermodynamic analysis. The possible reactions and the changes in Gibbs free energy are shown in Equations (3)–(11) and Figure 4.
Al_2_O_3_ + 2NaOH = 2NaAlO_2_ + H_2_O(3)
MgAl_2_O_4_ + 2NaOH = 2NaAlO_2_ + MgO + H_2_O(4)
Fe_2_O_3_ + 2NaOH = 2NaFeO_2_ + H_2_O(5)
2AlN + 2NaOH + 1.5O_2_ = 2NaAlO_2_ + H_2_O + N_2_(6)
Al_2_O_3_ + Na_2_CO_3_ = 2NaAlO_2_ + CO_2_(7)
MgAl_2_O_4_ + Na_2_CO_3_ = 2NaAlO_2_ + MgO + CO_2_(8)
Fe_2_O_3_ + Na_2_CO_3_ = 2NaFeO_2_ + CO_2_(9)
2AlN + Na_2_CO_3_ + 1.5O_2_ = 2NaAlO_2_ + CO_2_ + N_2_(10)
SiO_2_ + NaAlO_2_ = NaAlSiO_4_(11)

From the reaction curves (3)–(6), it can be seen that the Gibbs free energy of the reactions of Al_2_O_3_, MgAl_2_O_4_, Fe_2_O_3_ and AlN with NaOH were negative at room temperature, which indicates that the thermodynamic reaction conditions are sufficient. The Al_2_O_3_ of MgAl_2_O_4_ phase was combined with NaOH to obtain NaAlO_2_ and release MgO. Moreover, AlN was oxidized to generate non-toxic N_2_. The reaction curves (7)–(9) show that the Gibbs free energy values of the reactions were negative only at higher temperatures, which indicates that Na_2_CO_3_ was less active compared to NaOH. The reaction curve (10) shows that the thermodynamic conditions for the reaction of AlN and Na_2_CO_3_ to produce NaAlO_2_, CO_2_ and N_2_ were sufficient. From the reaction curve (11), it is clear that insoluble NaAlSiO_4_ was produced by the combination of SiO_2_ and NaAlO_2_, resulting in the loss of alumina in the SAD.

These results demonstrate that the reactions of the SAD with the two alkali additives (NaOH and Na_2_CO_3_) were feasible. All the Al-containing compounds (Al, Al_2_O_3_, AlN and MgAl_2_O_4_) in SAD can react with alkali additives to form readily soluble NaAlO_2_. In addition, the other impurities (Mg, Fe, and Si) generated insoluble species, thus achieving the separation and extraction of valuable aluminum elements from SAD.

### 3.2. Effects of Different Factors on the Roasting System

In order to ensure that the desired compounds are produced in the clinkers, the extent of the roasting of the mixed materials should be strictly controlled. Since both the alkali additives can thermodynamically react with SAD to produce the expected compounds, the effects of these alkali additives on the roasting system were examined.

The effects of different factors on the recoveries of Al and Na were studied by conducting a variety of experiments for each additive type, various roasting temperatures and ingredients depending on the n(N/A) (see Table 2 and Figure 5).

#### 3.2.1. Effects of Different Factors on the Recoveries of Al and Na

For the alkali additives, NaOH and Na_2_CO_3_, the recoveries of Al and Na in the clinkers under different temperatures are shown in Figure 5a and Figure 5b, respectively. As is evident from Figure 5a,b, the roasting temperature had a significant effect on the recoveries of Al and Na. With the increase in the roasting temperature, the recoveries of both the Al and Na initially increased and then decreased. When NaOH was used as the additive, the recoveries of Al and Na reached the maximum values of 83.47% and 89.06%, respectively, at the roasting temperature of 950 °C. However, when Na_2_CO_3_ was used as the additive, the highest recoveries of Al and Na had values of 80.90% and 88.72%, respectively, at the roasting temperature of 1100 °C. Additionally, the recoveries of Al and Na were similar in the within ±50 °C interval of the optimum temperature, which indicates that the range of the optimum temperature was wide and simple to control.

On the one hand, the mixed materials primarily relied on the solid−solid reaction during the roasting process. The generation of less liquid phase in the clinkers at low roasting temperatures led to slow reaction rates, large porosity and poor dissolution performance. On the other hand, the generation of a large amount of liquid phase in the clinkers at high roasting temperatures resulted in the rapid volatilization of the alkali, low porosity and poor dissolution performance [28]. Therefore, the appropriate liquid phase and porosity of clinkers are available only at a certain roasting temperature, which offers a better reaction rate and dissolution performance.

The n(N/A) is a critical parameter which affects the extraction of the valuable Al element from SAD. The n(N/A) represents the amount of alkali additives which have been added to the system. For the additives NaOH and Na_2_CO_3_, the recoveries of Al and Na in the clinkers under different n(N/A) are shown in Figure 5c and Figure 5d, respectively. As shown in Figure 5c, when the n(N/A) was increased from 1.0 to 1.2, the recovery of Al increased from 83.47% to 91.83%, whereas that of Na increased from 89.36% to 93.64%. When Na_2_CO_3_ was used as the additive, the n(N/A) of 1.3 was optimal for extracting NaAlO_2_, while the Al and Na recoveries were 90.79% and 92.03%, respectively. This could be due to the fact that the low quantities of alkali were not sufficient enough to adequately support the reaction of the aluminum-containing compounds (Al, Al_2_O_3_, AlN, and MgAl_2_O_4_) in SAD with Na_2_O to produce soluble NaAlO_2_, thus resulting in a poor dissolution performance of the clinkers. With the increase in the n(N/A), the recoveries of Al and Na gradually declined. This can be attributed to the production of some insoluble substances due to excessive amounts of alkali, which prevented the further dissolution of NaAlO_2_.

It is worth mentioning that the NaOH additive resulted in a lower roasting temperature and slightly higher Al and Na recoveries compared to the Na_2_CO_3_ additive under the optimal roasting conditions. However, due to the NaOH deliquescence, the mixed materials were able to easily absorb the moisture in the air during the experiment, thus causing the hydrolysis of the AlN in SAD and releasing a large amount of toxic ammonia. Additionally, the produced cylindrical samples quickly lost their strengths, forming into a slurry. Therefore, Na_2_CO_3_ was selected as the appropriate additive for the subsequent experiments.

#### 3.2.2. Effects of Different Factors on the Mineralogical Phases

The mineralogical analysis of the roasting clinkers and leaching residues at various roasting temperatures is illustrated in Figure 6. The main phases in the roasting clinkers included NaAlO_2_, MgO, NaFeO_2_ and NaAlSiO_4_, which was consistent with the results of the thermodynamic analysis. The diffraction peak intensity of NaAlO_2_ in the roasting clinkers significantly increased from 1000 °C to 1050 °C, which was the critical solid phase of the Al recovery. Moreover, SiO_2_ played a negative role in the recovery of Al because SiO_2_ was able to react with soluble NaAlO_2_ to produce insoluble NaAlSiO_4_.

After the leaching process, the NaAlO_2_ in the roasting clinkers was completely dissolved, and the NaFeO_2_ was hydrolyzed to generate insoluble Fe(OH)_3_ and release NaOH. Therefore, the main phases in the leaching residues were MgO, MgAl_2_O_4_, Fe(OH)_3_ and NaAlSiO_4_. The diffraction peak intensity of MgO in the leaching residues continued to increase from 1000 °C to 1150 °C, while the diffraction peak intensity of MgAl_2_O_4_ significantly decreased. This indicates that an increase in the temperature was beneficial to the reaction (8). A small amount of MgAl_2_O_4_ was still observed in the roasting clinkers and leaching residues due to the low amounts of Na_2_CO_3_. The phase composition of the roasting temperature above 1150 °C remained unchanged, indicating that the mineralogical phase of the roasting clinkers was more effective at 1150 °C, which is consistent with the results shown in Figure 5c.

The XRD patterns of the roasting clinkers and leaching residues at different n(N/A) are shown in Figure 7. As the n(N/A) increased, the diffraction peak intensities of NaAlO_2_ and MgO in the roasting clinkers continued to increase, while the peak intensity of MgAl_2_O_4_ continued to decrease. This shows that the reaction (8) was facilitated by increasing the n(N/A). When the n(N/A) exceeded the value of 1.2, the SiO_2_ disappeared and NaAlSiO_4_ continued to decrease, while Na_1.95_Al_1.95_Si_0.05_O_4_ was generated as a result of the reaction (12), and this was easily dissolved in the alkali solution [29].
0.1SiO_2_ + 1.95Na_2_CO_3_ + 1.95Al_2_O_3_ = 2Na_1.95_Al_1.95_Si_0.05_O_4_ + 1.95CO_2_
(12)

With the increase in the n(N/A) from 1.0 to 1.3, the MgO in the leaching residues significantly increased, while the MgAl_2_O_4_ continued to decrease, which is consistent with the XRD results of the roasting clinkers. The phase composition of the system with an n(N/A) of over 1.3 was essentially the same as that of the system with a lower molar ratio, indicating that the mineralogical phases of the roasting clinkers with the n(N/A) of 1.3 were more effective, which is consistent with the results shown in Figure 5d.

### 3.3. Effects of Different Factors on the Leaching System

Since the structure of solid NaAlO_2_ is different from the structure of the aluminate ions in the solution, the dissolution of NaAlO_2_ in the roasting clinkers is actually a chemical reaction, as given by the reaction (13) [30].
(13)$${\mathrm{Na}}_{2}O\cdot {\mathrm{Al}}_{2}O_{3}\left(s\right){+4H}_{2}{O=\mathrm{Na}}^{+}{+2\mathrm{Al}(\mathrm{OH})}_{4}^{-}$$

In this section, the effects of various leaching factors including the leaching temperature, leaching time and liquid-to-solid ratio on the recovery of Al were evaluated. These experiments were conducted at the roasting temperature of 1150 °C, n(N/A) of 1.3 and roasting time of 1 h.

#### 3.3.1. Effects of the Leaching Temperature

Figure 8a shows the recovery efficiency of the Al element in the roasting clinkers at different leaching temperatures as a function of the leaching time. The reaction was severe within the first 15 min, and the NaAlO_2_ in the roasting clinkers was continuously dissolved. However, no significant increase in the recovery of the Al was observed with the further extension of the time. When the leaching temperature was increased from 30 °C to 90 °C, the 15 min Al recovery increased from 68.79% to 88.11%. The leaching temperature had a significant impact on the recovery of the Al, because the increase in the temperature was beneficial in accelerating the mass transfer and diffusion.

#### 3.3.2. Effect of the Liquid-to-Solid (L/S) Ratio

The effect of L/S ratio within the range of 5–15 mL·g^−1^ on the leaching process was investigated at a leaching temperature of 60 °C. As shown in Figure 8b, when the L/S ratio was increased from 5 mL·g^−1^ to 10 mL·g^−1^, the recovery of the Al increased significantly. However, when the L/S ratio was increased to 15 mL·g^−1^, the recovery of the Al became relatively stable. Therefore, the L/S ratio of the leaching process had a significant influence on the recovery of the Al within a certain range. Increasing the amount of leaching solution can help to reduce the slurry viscosity and increase the concentration difference of the aluminate ions on the solid–liquid interface, thus accelerating the dissolution of NaAlO_2_ in the roasting clinkers. Therefore, an L/S ratio of 10 mL·g^−1^ was selected as the optimum leaching condition for further experimentation.

### 3.4. Analysis of the Leaching Kinetics

The leaching process of the NaAlO_2_ in the roasting clinkers is a liquid-to-solid heterogeneous reaction system, which takes place at the phase interfaces. In order to obtain a more efficient process, the kinetics of the dissolution of NaAlO_2_, including the reaction rate constant, reaction order, activation energy and the rate-determining step, should be thoroughly evaluated. For the NaAlO_2_ dissolution in the roasting clinkers, the rate of reaction is given by Equation (14) [29].
(14)$$-\frac{{dC}_{Al}}{dt}{=k\;C}_{Al}{}^{n}$$
where ${\;C}_{Al}$ is the content of NaAlO2 in the roasting clinkers after leaching for time *t*, *t* is the leaching time, *n* is the reaction order and *k* is the reaction rate constant, s^−1^.

The experimental results shown in Figure 8a were used to plot the different reaction orders (*n* = 0, 1, and 2) in Equation (14) and the corresponding results are shown in Figure 8c–e), respectively. The confidence level of the fitted straight line was high for the first order reaction (*n* = 1), while the linear relationship exhibited strong agreement in the second order reaction (*n* = 2), except for a slight deviation at 363.15 K. It has been shown that many reactions can be satisfactorily correlated using both the first and second order reactions. Therefore, it was not prudent to rely only on the linear correlation of the reaction equation to determine the reaction order. The half-life method was used to effectively determine the reaction order of the dissolution of NaAlO_2_ in the roasting clinkers (Equation (15)):(15)$$\chi_{Al}=\frac{C_{Al,\;0}-C_{Al}}{C_{Al,\;0}}$$

The logarithmic form of Equation (15) is expressed as Equation (16):(16)$$k=\frac{1}{t}\;\ln\;\frac{1}{{1\;-\;\chi }_{Al}}$$
where $\chi_{Al}$ is the dissolution rate of NaAlO_2_ in the roasting clinkers at the time *t*, and $C_{Al,\;0}$ is the initial NaAlO_2_ content in the roasting clinkers.

When the dissolution rate of NaAlO_2_ was 50% ($\chi_{Al}$ = 0.5), the half-life of the first order reaction $t_{1/2}$ is given by Equation (17), whereas the half-life of the second order reaction $t_{1/2}^{\prime }$ is written as Equation (18):(17)$$t_{1/2}=\frac{\ln2}{k}$$
(18)$$t_{1/2}^{\prime }=\frac{1}{{k\;C}_{Al,\;0}}$$

The half-life results for first and second order reactions at different leaching temperatures are shown in Figure 8f. When the reaction order was one (n = 1), the half-life of the dissolution of NaAlO_2_ within the leaching temperature of 303.15–363.15 K was 7.67–4.05 min. However, when the reaction order was two (n = 2), the half-life within the temperature of 303.15–363.15 K was 6.15–1.82 min. Combined with the experimental results shown in Figure 8a, the half-life showed high reliability in first order reaction, indicating that the dissolution of the NaAlO_2_ was proportional to the reactant concentration.

The temperature dependence of the leaching reaction rate constant is expressed using the Arrhenius correlation (Equation (19)):(19)$$k=A\;\cdot {\;\exp(-E}_{a}/RT)$$
where *k* is the overall reaction constant, A is the pre-exponential factor (min^−1^), R is the ideal gas constant (8.3145 J·mol^−1^·K^−1^) and $E_{a}$ (J·mol^−1^) is the apparent activation energy of the dissolution of NaAlO_2_. The Arrhenius correlation can be rewritten in logarithmic form as Equation (20):(20)$$\ln k=\mathrm{lnA}\;-\;\frac{E_{a}}{RT}$$

In general, when the activation energy is 40–300 kJ·mol^−1^, the reaction is controlled by the interface. However, when the activation energy is 8–20 kJ·mol^−1^, the reaction is controlled by the diffusion process [30]. Finally, the activation energy ($E_{a}$) calculated from the slope of Figure 9 was found to be 9.69 kJ·mol^−1^, which is lower than the value of 20 kJ·mol^−1^, thus confirming that the dissolution of NaAlO_2_ was controlled by the diffusion process. This result is also similar to the result of the dissolution of NaAlO_2_ mentioned in He’s study, where the activation energy was reported to be 11.4010 kJ·mol^−1^ [30].

### 3.5. Thermal Analysis and Product Characterization

In order to further evaluate the phase transition of the roasting process, the thermogravimetric and differential scanning calorimetry (TG-DSC) curves of the mixed materials of SAD and Na_2_CO_3_ (n(N/A) of 1.3) at a heating rate of 10 °C·min^−1^ are shown in Figure 10.

The weight loss of the mixed materials can be divided into three stages: (1) from room temperature to 550 °C, the weight loss was caused by the evaporation of the attached water, and the weight was reduced by 1.24%; (2) from 550 °C to 1050 °C, CO_2_ was released due to the reaction of Na_2_CO_3_ with the components in the SAD, resulting in a weight loss of 8.78 wt.%, which is also supported by the fact that Na_2_CO_3_ cannot undergo thermal decomposition within the temperature range of 550–1050 °C; (3) from 1050 °C to 1200 °C, the weight of the sample remained basically unchanged, indicating that the main reaction was completed.

The DSC curve showed three obvious endothermic peaks at 564 °C, 686 °C and 820 °C in the heating process of the mixed materials. Additionally, two obvious exothermic peaks at 904 °C and 1052 °C were observed. The enthalpy changes in the main reactions during the roasting process (see Section 3.1) are presented in Table 3, which indicate that the reactions (7)–(9) were endothermic, while the reactions (10) and (11) were exothermic. Previous studies have shown that Al_2_O_3_ and Na_2_CO_3_ react mainly at 500–700 °C, whereas Fe_2_O_3_ reacts with Na_2_CO_3_ at about 850 °C. Combining the thermodynamics calculations, XRD results and the TG-DSC analysis, the endothermic peaks at 564 °C, 686 °C and 820 °C should be attributed to reactions (7), (8) and (9), respectively. In addition, the exothermic peaks at 904 °C and 1052 °C corresponded to reactions (11) and (10), respectively, which were determined using the changes in enthalpy. The main chemical reactions of the roasting process were completed at around 1150 °C, which was consistent with the optimal roasting temperature derived in Section 3.2.

The micromorphology of the roasting clinkers and leaching residues with the n(N/A) of 1.3 were observed using a scanning electron microscope (SEM). The energy dispersive spectroscopy (EDS) images are shown in Figure 11e–h. Figure 11a,b shows that the particles in the roasting clinkers were clustered and had a particle size of about 10–30 μm. The shape of the NaAlO_2_ particles was smooth and irregular. The morphology of the MgO particles was agglomerated and spherical, with a particle size of 2–4 μm. Figure 11c shows that the NaAlSiO_4_ in the leaching residues had a smooth block-shaped appearance, while MgAl_2_O_4_ was octahedral, as shown in Figure 11d.

The chemical compositions of the roasting clinkers and leaching residues under optimal conditions (roasting temperature of 1150 °C, Na_2_CO_3_/Al_2_O_3_ molar ratio of 1.3, roasting time of 1 h, leaching temperature of 90 °C, L/S ratio of 10 mL·g^−1^ and leaching time of 30 min) are presented in Table 4. The contents of N and Cl in the roasting clinkers reduced to 0.072 wt.% and 0.12 wt.%, respectively. Moreover, the AlN was oxidized to produce harmless N_2_, and the chloride was evaporated to the gaseous phase, which was recovered after the condensation. The removals of N and Cl reached 98.93% and 97.14%, respectively. Therefore, the green toxification of SAD can be achieved in the roasting process without any pretreatment.

## 4. Conclusions

SAD is a hazardous solid waste as well as an alumina-rich resource. The safe disposal and cost-effective recycling of SAD are serious challenges. A combined dry pressing and alkali roasting process was developed for the sustainable extraction of the valuable Al element from SAD. Based upon the results, the following conclusions can be drawn:Both alkali additives (NaOH and Na_2_CO_3_) could react efficiently with SAD to obtain high recoveries of Al and Na. However, due to the deliquescence of the NaOH, the AlN reacted with water to rapidly release large amounts of NH_3_. Therefore, the NaOH additive was not suitable for the dry pressing step.The molar ratio of Na_2_CO_3_/Al_2_O_3_ and the roasting temperatures significantly affected the phase composition and leaching performance of the roasted products. Under optimal conditions, the recoveries of the Al and Na in the roasting clinkers reached the values of 90.79% and 92.03%, respectively.The removal efficiencies of the N and Cl reached the values of 98.93% and 97.14%, respectively. The green detoxification and efficient extraction of the valuable Al from the SAD were simultaneously achieved in the roasting process. This process has the sustainable developing and practical application on SAD recovery.The leaching kinetics showed that the NaAlO_2_ dissolution in the roasting clinkers was a first order reaction and controlled by a layer diffusion process. The apparent activation energy was 9.69 kJ·mol^−1^. The experiments of desilication and crystal seed decomposition can be conducted on synthesized metallurgical grade alumina in future.

## Figures and Tables

**Figure 1 materials-15-05686-f001:**
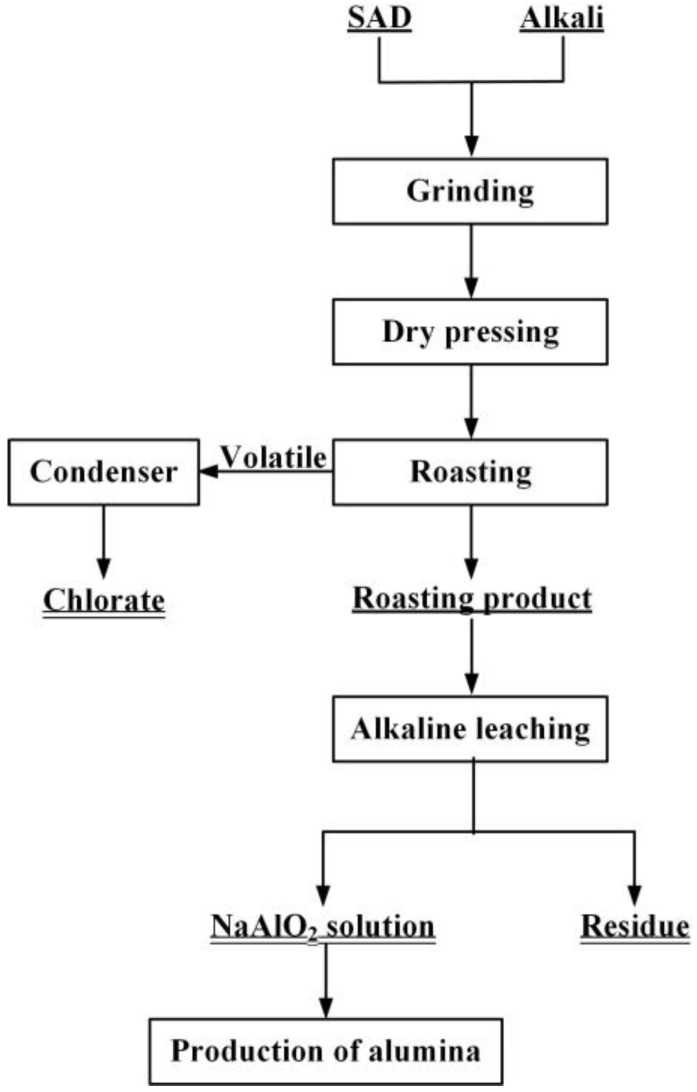
Flowchart of the preparation of alumina using SAD as the raw material.

**Figure 2 materials-15-05686-f002:**
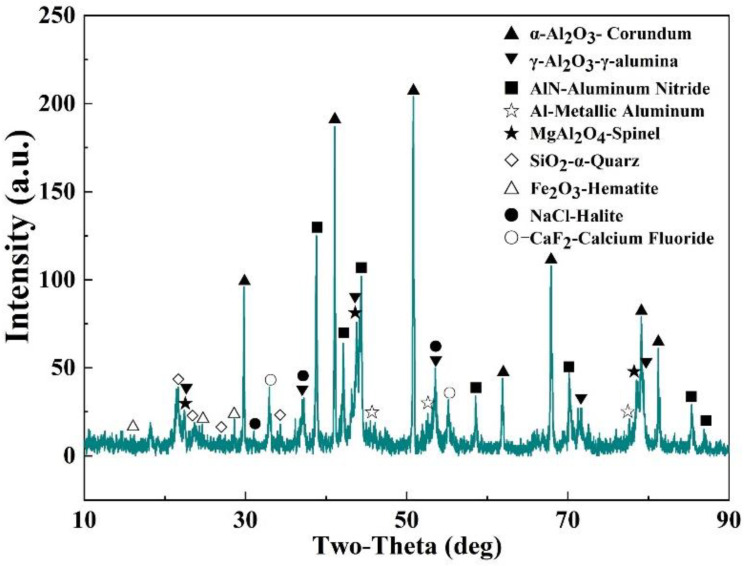
XRD pattern of the SAD sample.

**Figure 3 materials-15-05686-f003:**
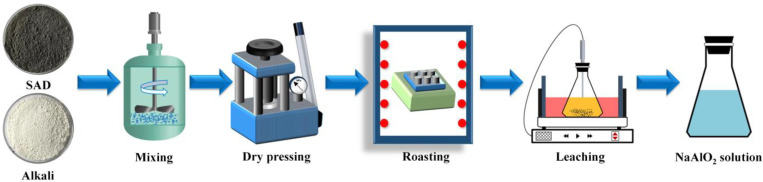
Schematic of the experimental setup and procedure.

**Figure 4 materials-15-05686-f004:**
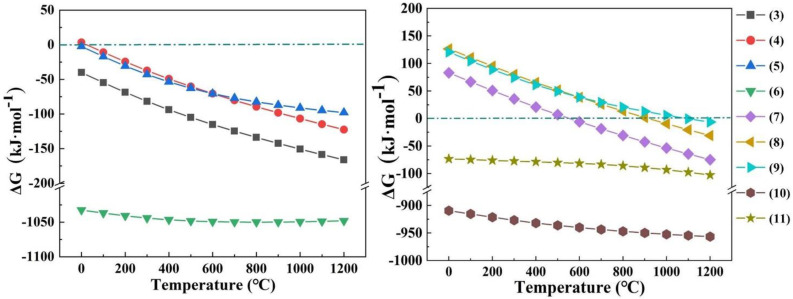
Relationship of the changes in Gibbs free energy with the temperature for the reactions during the roasting process.

**Figure 5 materials-15-05686-f005:**
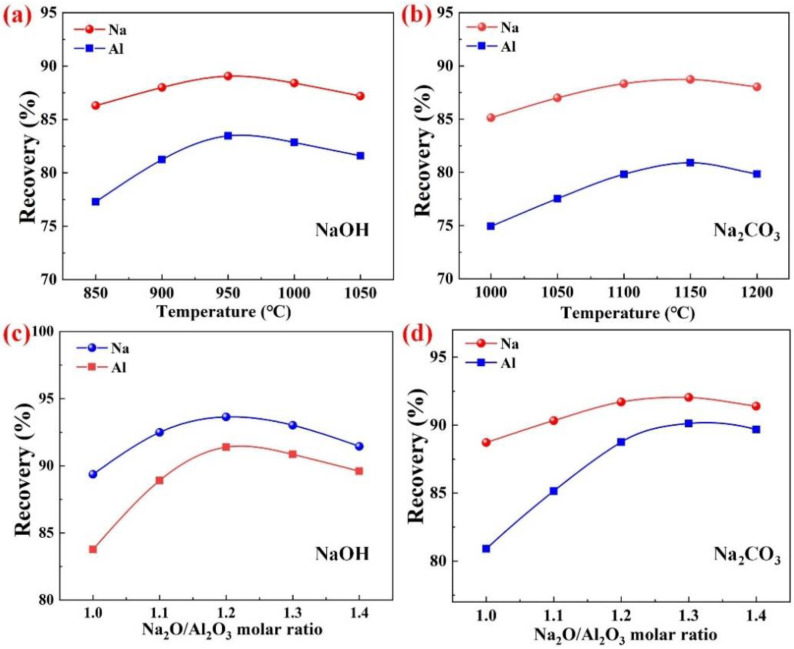
Recovery of Al and Na in roasting clinkers under different roasting conditions: roasting temperature with NaOH additive (**a**), n(N/A) with NaOH additive (**b**), roasting temperature with Na_2_CO_3_ additive (**c**), and n(N/A) with Na_2_CO_3_ additive (**d**).

**Figure 6 materials-15-05686-f006:**
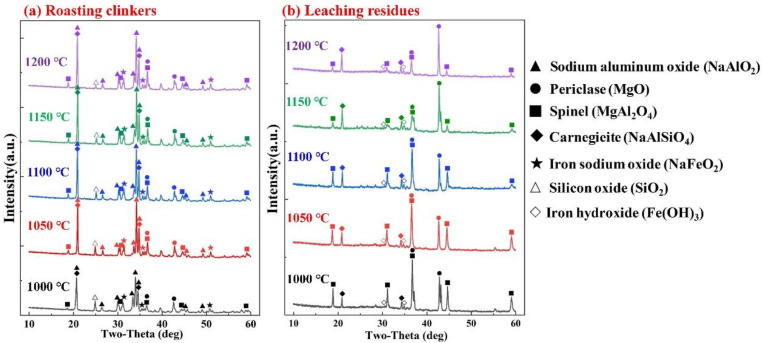
XRD patterns of the roasting clinkers (**a**) and leaching residues (**b**) under different roasting temperatures.

**Figure 7 materials-15-05686-f007:**
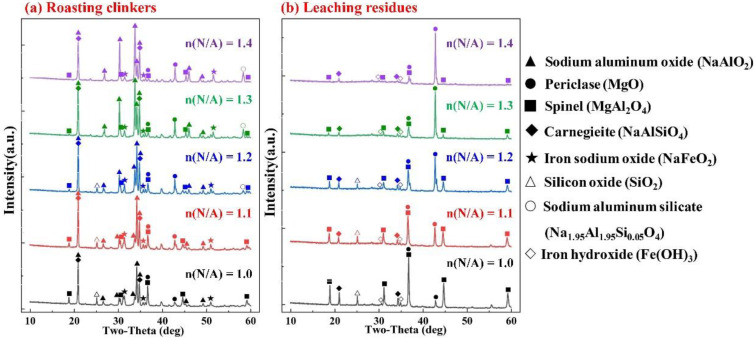
XRD patterns of the roasting clinkers (**a**) and leaching residues (**b**) under different Na_2_CO_3_/SAD mass ratios.

**Figure 8 materials-15-05686-f008:**
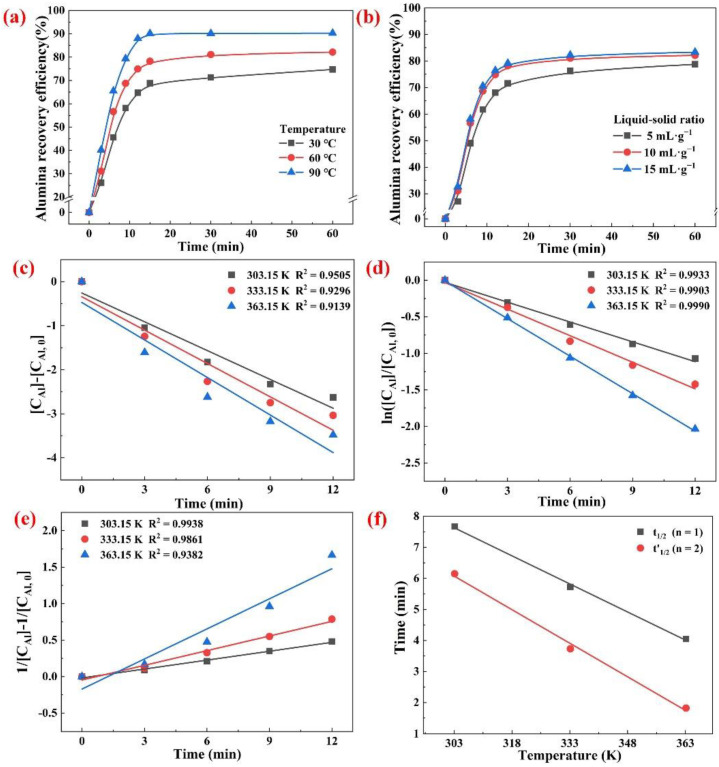
Recovery of Al in the roasting clinkers under different leaching temperatures (**a**) and L/S ratios (**b**), with fitting experimental data for different reaction orders: *n* = 0 (**c**), *n* = 1 (**d**) and *n* = 2 (**e**), and the half-life of leaching process at *n* = 1 and *n* = 2 (**f**).

**Figure 9 materials-15-05686-f009:**
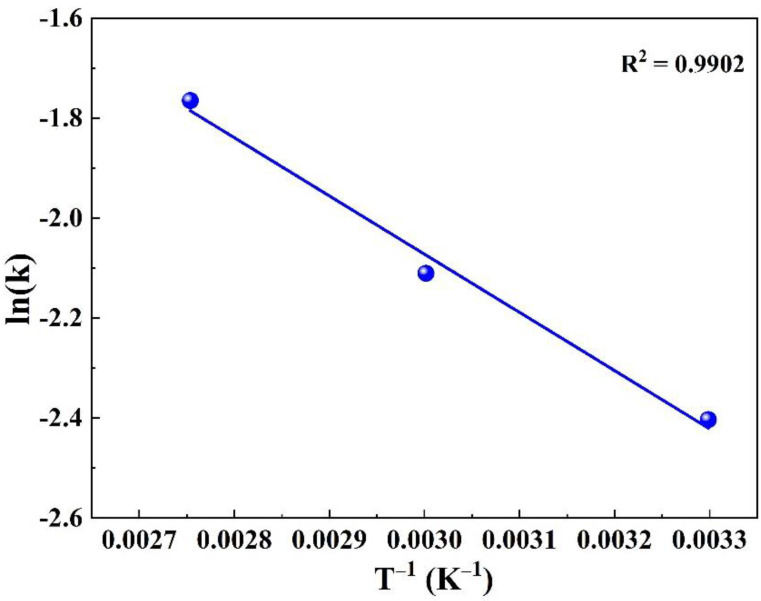
Relationship between the leaching rate constant *k* and temperature *T*.

**Figure 10 materials-15-05686-f010:**
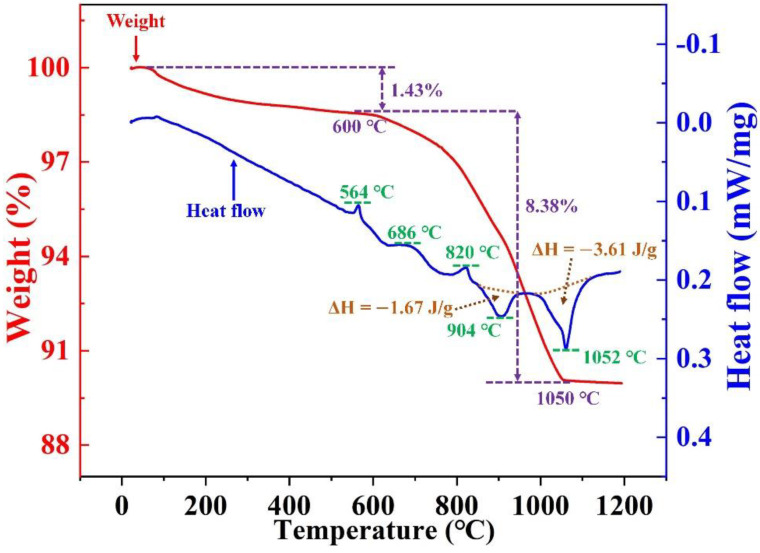
TG-DSC curves of SAD–Na_2_CO_3_ mixtures for the n(N/A) of 1.3.

**Figure 11 materials-15-05686-f011:**
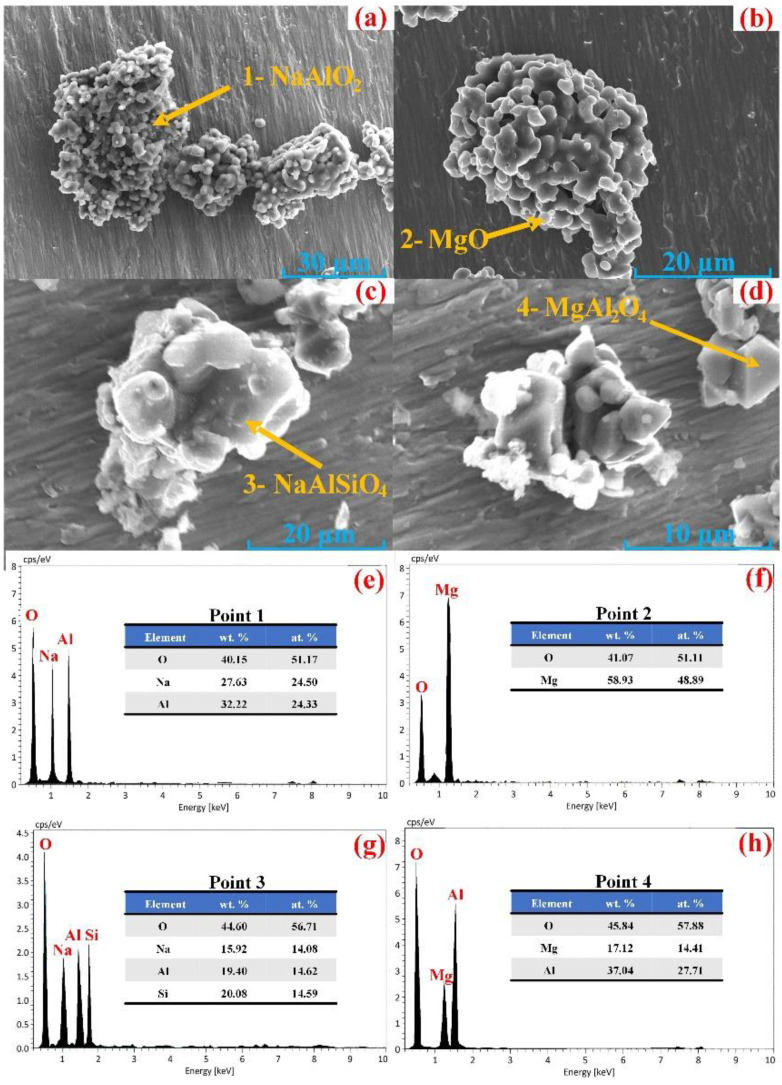
SEM micrographs of the roasting clinkers (**a**,**b**), leaching residues (**c**,**d**) and EDS (**e**–**h**).

**Table 1 materials-15-05686-t001:** Chemical composition of SAD.

Element	Al	O	N	F	Na	Ca	Si	Mg	Fe	K	Cl
Content (wt.%)	38.23	38.14	6.7	0.25	2.16	0.32	2.04	6.18	1.04	0.57	4.2

**Table 2 materials-15-05686-t002:** List of roasting parameters studied in the experiments.

Parameters	Parameter Values	Fixed Roasting Parameters	Fixed Leaching Parameters
Roasting temperature (°C)	850, 900, 950, 1000, 1050	Additive of NaOH, t = 1 h, n (N/A) = 1.	C(Na_2_O) = 60 g/L, α_k_ = 3.0, T = 90 °C, t = 60 min
n(N/A)	1.0, 1.1, 1.2, 1.3, 1.4	Additive of NaOH, T = 950 °C. t = 1 h.	
Roasting temperature (°C)	1000, 1050, 1100, 1150, 1200	Additive of Na_2_CO_3_, t = 1 h, n (N/A) = 1.	
n (N/A)	1.0, 1.1, 1.2, 1.3, 1.4	Additive of Na_2_CO_3_, T = 1100 °C, t = 1 h.	

**Table 3 materials-15-05686-t003:** Enthalpy changes in the reactions during the roasting process.

Reactions	ΔH at 500 °C (kJ·mol^−1^)	ΔH at 1000 °C (kJ·mol^−1^)
Al_2_O_3_ + Na_2_CO_3_ → 2NaAlO_2_ + CO_2_	110.00	85.90
MgAl_2_O_4_ + Na_2_CO_3_ → 2NaAlO_2_ + MgO + CO_2_	154.58	132.19
Fe_2_O_3_ + Na_2_CO_3_ → 2NaFeO_2_ + CO_2_	135.38	90.93
AlN + 0.5Na_2_CO_3_ + 0.75O_2_ → NaAlO_2_ + 0.5CO_2_ + 0.5N_2_	−452.605	−461.11
SiO_2_ + NaAlO_2_ → NaAlSiO_4_	−69.95	−38.98

**Table 4 materials-15-05686-t004:** Compositions of the roasting clinkers and the leaching residues under optimal conditions (%).

Roasting Clinkers	Al	O	F	Na	Ca	Si	Mg	Fe	N	K	Cl
Content (wt.%)	27.06	34.88	0.19	29.67	0.24	1.55	4.98	0.79	0.072	0.19	0.12
Leaching residues	Al	O	F	Na	Ca	Si	Mg	Fe	N	K	Cl
Content (wt.%)	10.64	43.63	0.87	10.10	1.36	7.52	21.27	4.26	0.04	0.06	0.03

## Data Availability

Not applicable.

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
