# Peer review of "Effective Extraction of the Al Element from Secondary Aluminum Dross Using a Combined Dry Pressing and Alkaline Roasting Process"

_materials, 2022, doi:10.3390/ma15165686_

Round 1

Reviewer 1 Report

After a deep reviewing of this paper, I feel it is suitable for publication in Materials in the present form. I am suggesting no changes in the entire paper, so it is a good effort for the treatment of aluminum production of solid wastes.

The conclusions are adequate and this is a paper mostly important to the readers of the whole aluminum industry topics.

Reviewer 2 Report

The manuscript contains some interesting results. However, before manuscript can be considered for publication, it should be reviewed in accordance with the following comments:

 Abstract: is adequate

1. Introduction:

- the last 3 paragraphs should be moved to Section 2. Materials and Methods

2. Materials and Methods

- In the SAD characterization step, what were the XRD parameters? material condition?

- how was chemical composition determined?

- in Experimental Procedure, preset temperatures should be cited

- why was a corundum crucible used for roasting?

- why were specimens kept for 1h - roasting time?

- what is “For mineralogical study”? what does it mean?

- how were samples for SEM analysis prepared? and the used parameters?

- how were the alkalis and roasting temperatures selected? DOE (design of experiments)?

3. Results and Discussion

- why were 2 alkalis and specimens kept for 10h at 410C after solidification?

- how were the phases identified in XRD patterns (Figs 6 and 7)?

- references for Eqs. 3-11 and 3-12 should be presented

- labels in images of Fig. 11 should be improved

- what is the accuracy/precision of EDS for determining chemical compositions of clinkers and residues?

- the authors state that “The removals of N and Cl reached 98.93% and 97.14%”. Please, elucidate this issue.

4. Conclusions

Readers expect a sharper focus on the most important aspects of the work. Conclusions should not simply repeat the results and discussion. They should list concisely and clearly what new knowledge has been contributed to materials.

Reviewer 3 Report

Excellent paper. I would have liked to see the precipitation of the alumina at the end of the process to see its quality. Would the alumina have a high enough quality for electrolysis? Or for tabular alumina? I would also have liked an economical analysis. I would recommend to remove the notion that this process is economical because there are no analysis presented or scale up proposition.
